# Experimental and Statistical Study on Mechanical Characteristics of Geopolymer Concrete

**DOI:** 10.3390/ma13071651

**Published:** 2020-04-02

**Authors:** Yifei Cui, Kaikai Gao, Peng Zhang

**Affiliations:** Center for Durability & Sustainability Studies of Shandong Province, Qingdao University of Technology, Qingdao 266033, China; cuiyifei@qut.edu.cn (Y.C.); gaokaikai0524@126.com (K.G.)

**Keywords:** geopolymer concrete, statistics, mechanical characteristics

## Abstract

This paper studies the statistical correlation in mechanical characteristics of class F fly ash based geopolymer concrete (CFGPC). Experimentally measured values of the compressive strength, elastic modulus and indirect tensile strength of CFGPC specimens made from class F fly ash (CFA) were presented and analyzed. The results were compared with those of corresponding ordinary Portland cement concrete (OPCC) using statistical hypothesis tests. Results illustrated that when possessing similar compressive and tensile strength, the elastic modulus for CFGPC is significantly lower than that of OPCC. The corresponding expressions recommended by standards for the case of OPCC is proved to be inaccurate when applied in the case of CFGPC. Statistical regression was used to identify tendencies and correlations within the mechanical characteristics of CFGPC, as well as the empirical equations for predicting tensile strength and elastic modulus of CFGPC from its compressive strength values. In conclusion, CFGPC and OPCC has significant differences in terms of the correlations between mechanical properties. The empirical equations obtained in this study could provide relatively accurate predictions on the mechanical behavior of CFGPC.

## 1. Introduction

Ordinary Portland cement (OPC) is the most widely used concrete binder. OPC, aggregates and water together form ordinary Portland cement concrete (OPCC) that dominates the market of construction materials [1]. According to the USGS Mineral Program Cement Report [2], the world OPC production was over 3800 million tons in 2018. The tremendous amount of OPC use is of great concern given that the production of OPC results in serious greenhouse gas emissions. It has been speculated that carbon dioxide (CO_2_) released from fossil fuel combustion in cement manufacture could be the primary anthropogenic driver of climate change [3]. Besides, the durability problems caused by OPC [4,5] are of great concern in terms of directly damaging the safety of structures [6]. 

In this background, geopolymers have attracted much attention as a sustainable binder with outstanding environmental advantages, good mechanical properties and reliable durability [7]. The term “geopolymer” describes a family of the synthetic polymer binders that have a pozzolan–zeolite silicon–oxygen–aluminum porous structure [8,9]. The synthesis process of geopolymers is called “geopolymerization”, which consists of dissolution, transportation or orientation, and polycondensation in an exothermic process [7,10]. This binder is environmentally friendly as it uses the industry wastes such as the class F fly ash as raw material. Class F fly ash is an important by-product generated from power generation industries. When class F fly ash is activated by alkaline activators, the glassy structures are transformed into amorphous composites that possess similar mechanical characteristics as OPC [11]. 

The fresh and hardened class F fly ash based geopolymer concrete (CFGPC) has many differences with fresh and hardened OPCC. The fresh CFGPC is a sticky paste with high viscosity and low fluidity. The traditional ways for testing the slump of concrete are not suitable to measure the workability of CFGPC [9]. For one thing, CFGPC shows high adhesion with steel and tends to stick to the steel slump core during the tests; for another, although the mobility of the fresh CFGPC is restrained by the high viscosity, the high water-retaining property and the good response to vibration make it actually very easy to be finished in the lab and on site [8]. The hardened CFGPC showed a smooth surface with high stiffness, which contributed by the amorphous products generated from geopolymerization. Besides the hard surface, the amorphous structure of CFGPC also provides it with many outstanding characteristics in resisting acid [12] and sulphate attack [13], as well as chloride penetration resistance [14] and fire resistance [10,15,16] compared with OPCC [17].

The development of CFGPC presents direct challenges to OPCC as leading construction material. Since the 1980s, the research on mechanical properties of CFGP paste and mortars has been very encouraging [18]; however, published research has been mainly focused on the mix design [19], manufacturing process [20], curing conditions [21], the process of geopolymerization and the chemical structures of products [7]. To date, the relation between the crucial mechanical characteristic of GPCC is still undefined. Unlike OPCC, there is no widely accepted correlations in any standards to predict tensile strength and elastic modulus from specific compressive strength. 

As part of research committed to study the potential of CFGPC as a future construction material, this study aims to establish relationships between key mechanical characteristics of CFGPC based on statistical studies that include using hypothesis tests and statistical regression. The experimentally determined compressive strength, splitting tensile strength and elastic modulus of CFGPC mix were analyzed for suggesting certain correlations between compressive and tensile strength or elastic modulus.

## 2. Experiments 

The experimental program in this research aimed to test CFGPC and corresponding OPCC samples for compressive strength, indirect tensile strength and elastic modulus. The binder material of CFGPC was alkaline activated class F fly ash geopolymer, while that of OPCC was hydrated general purpose ordinary Portland cement.

### 2.1. Fly Ash

The fly ash used in this study is ASTM Class F fly ash from the Eraring thermal power station in New South Wales, Australia. Three batches of fly ash were used that are respectively referred to here as “CFA1”, “CFA2”and “CFA3”. The X-ray fluorescence (XRF) and loss on ignition (LOI) results obtained for these batches of fly ash are presented in Table 1.

### 2.2. Alkaline Liquid

Two different types of alkaline solutions were used in the casting of CFGPC. These were a 12 mol solution of NaOH (361 g of NaOH flakes per liter [19]) and grade D Na_2_SiO_3_. The 12 mol NaOH solution was prepared by mixing NaOH flakes (>99% purity, supplied by Redox Pty Ltd., Minto, NSW, Australia) with deionized water. The grade D Na_2_SiO_3_ solution contained 29.4 wt.% of SiO_2_ and 14.7 wt.% of Na_2_O.

### 2.3. Mix Design

The mix design of CFGPC is presented in Table 2. 

In this mix design, the SiO_2_/Al_2_O_3_ ratio of alkaline solution was kept at 2.5 [18,19,23], the alkaline liquids/fly ash ratio was 0.5 and the free water to fly ash ratio was 0.074. Three sizes of crushed granite coarse aggregates (Holcim Pty Ltd., Fyshwick, ACT, Australia) (7, 10 and 14 mm) as well as river sand (Holcim Pty Ltd., Fyshwick, ACT, Australia) were used, with the fine aggregates proportion of 36.6% by mass of the total aggregates. Two commercially available admixtures, a high-performance water reducer (CENTROX^TM^ MWR, Centriair Pty Ltd., Sydney, Australia) and a viscosity modifying agent (CENTROX^TM^ VM, Centriair Pty Ltd., Sydney, Australia) were used with a dosage of 0.9 L/kg fly ash [19], respectively. Both were used at the dosage rate of 900 mL per 100 kg of solid binder material, which was the fly ash in this case. 

The corresponding OPCC mix design is presented in Table 3, with the targeted compressive strength being the same as that of CFGPC (35 MPa).

### 2.4. Specimens

Specimens were cast in the form of standard 100 mm diameter by 200 mm height cylinders. A total number of 108 CFGPC and 108 OPCC cylinders were tested. The tests were compression tests [24], splitting tension tests [25] and elastic modulus tests [26], all performed according to the corresponding Australian Standards.

### 2.5. Casting

The OPCC samples were prepared according to ASTM C192 [27]. The manufacturing and curing processes for the GPC specimens were designed according to earlier research [9,28]. The alkali solution was prepared for 24 h in an environmental control room (23 ± 1 °C and 50% relative humidity) prior to casting to get a stable temperature. Coarse and fine aggregates in saturated surface dry (SSD) conditions were first well mixed in a laboratory concrete mixer for two minutes. The fly ash was poured slowly into the mixer for a period of two minutes of dry mixing. The alkaline solutions were then added into the blend and mixed for two more minutes. Finally, the tap water and additives were added during a final period of two minutes of mixing before the blend was poured into molds. 

### 2.6. Curing

The CFGPC and OPCC cylinders were covered with plastic lids immediately after casting to minimize moisture loss. The prepared specimens were kept in the environmental control room (ER, 50% humidity, 23 °C) for 24 h as it has been reported that a period of “resting time” before heating is good for the strength development [8,9,11]. The specimens, along with their molds and covers, were then subjected to heat curing in an electric oven at 80 °C for periods of 24 h. After heating, specimens were taken out of the molds and stored in the ER until testing. The OPCC cylinders were demolded after 24 h and cured in a fog room (100% humidity, 23 °C) for 28 days.

### 2.7. Testing

The heat cured CFGPC can achieve a similar strength development within 7 days as those of OPC after 28 days [8,9,11]; hence, all the CFGPC samples were tested on the 7th day after casting, while all the OPCC samples were tested on the 28th day. The compressive, tensile and elastic modulus tests were performed using a TECNOTEST^TM^ K300 ICUT, (Tecnotest Pty Ltd., Treviolo, Italy) testing machine according to the standard procedures described in AS1012.9 [24]. The indirect tensile strength (splitting tensile strength) of the cylinders was measured following AS1012.10 [25]. The tests for elastic modulus were carried out following Australian Standards AS1012.17 [26]. 

## 3. Results and Discussion

The test results were recorded by the data acquisition system and summarized for data analysis. As only a part of the data is reported here, details of full experimental results may be sourced from Cui [29]. The experimental data for CFGPC and OPCC were compared so as to determine any statistical significances between them. Then the data of CFGPC were subjected to regression analysis to identify the possible correlations.

### 3.1. Summary of the Mechanical Properties 

A summary of the mechanical properties of each of the CFGPC and OPCC mixes is presented in Table 4 and Table 5 respectively. 

The results in these tables represent the mean of each group. At least three identical cylinders from each mix were tested in each group, and their average strength was reported to the nearest 0.01 MPa or GPa. The density of samples was calculated in accordance with the requirements of Australia standard AS1012.12.1 [30] and recorded to the nearest 0.01 g/cm^3^.

As shown in Table 4, depending on the variations in fly ash, the 7-day compressive strength (f_c_^’^) of CFGPC samples varied within a range between 28.99 MPa and 46.18 MPa. For samples with different compressive strengths, the strength on splitting (f_t_) ranged from 2.66 MPa to 4.19 MPa, which makes an average 9.69% of the compressive strength. The elastic modulus (E_c_) of CFGPC varied with variation of the compressive strength, from 16.74 GPa to 24.29 GPa. 

The data in Table 4 and Table 5 show the corresponding mechanical testing results of CFGPC and OPCC, respectively. For OPCC samples, the average strength at splitting was approximately 9.01% of their compressive strength, which was slightly lower than that of CFGPC (9.69%). At a typical compressive strength value, the value of the elastic modulus of OPCC was much higher than the elastic modulus values obtained for CFGPC. Specifically, when the average compressive strength of CFGPC was 37.35 MPa and that of OPCC was 36.91 MPa, the average elastic modulus of CFGPC was only18.97 GPa, which was significantly (32.39%) lower than that of OPCC (28.06 GPa). The differences of mechanical characteristics between CFGPC and OPCC were analyzed using statistical hypothesis testing reported in the following section. 

### 3.2. Comparison of Microstructures of CFGPC 

Figure 1 shows the SEM photos for CFGPC samples made from the three different batches of fly ash: CFA1, CFA2 and CFA3. 

It explains why this experimental work obtained different strength from the same mix design. It is clear that the samples made from CFA2 had the lowest degree of unreacted fly ash spheres, which contributed to the highest average compressive strength of the three. The compressive tests of samples made from these three were about 35 MPa (CFA1), 42 MPa (CFA2) and 30 MPa (CFA3).

It has been reported that the total reacted SiO_2_ to Al_2_O_3_ ratio is crucial to the rate of polymerization in a CFGPC system [11,18]. Specific to this research, testing results showed that CFA2 had a better reaction rate and a SiO_2_/Al_2_O_3_ ratio in XRF close to 2.8. The samples made from this batch tended to achieve a better polymerization process and higher strength. Despite the special high compressive strength achieved from CFA2, the CFGPC samples manufactured from all the three batches of fly ash in this study showed similar correlations between compressive strength and tensile strength/elastic modulus. The statistical methods involved to express and compare this correlation are described in the following sections.

### 3.3. Comparison of fc’, ft and Ec of CFGPC and OPCC Specimens

The values of compressive strength, indirect tensile strength and elastic modulus of CFGPC and OPCC specimens were compared using the statistical F-test and T-test, and the results are shown in Table 6. 

In Table 6, the measurement used in the T-test is called the t value. The t value is a ratio, which is usually denoted by t_s_. The numerator of this ratio is the difference between the mean values of samples in groups 1 and 2, respectively [31]. 

At a certain significance level, the T-test is conducted by comparing the t static (t_s_) with a given value called the t critical, the value of which can be read from the t-distribution critical values table, which is usually abbreviated to t-Table [32]. In this study, as the significance level α was determined to be 0.05, the t critical was denoted by t_0.05_. When the magnitude α is set as 0.05, if the t_s_ is smaller than the t 0.05 with a probability (P) larger than 0.05, the null hypothesis is supported with at least 95% confidence; otherwise, the study hypothesis is supported with at least 95% confidence. If the null hypothesis is supported when α = 0.05, there is no significant difference between these two means in the 95% confidence interval; otherwise, there is a significant difference between them in the 95% confidence interval.

Usually, the T-test is based on the assumption of homogeneous variances in the two groups being compared; otherwise a T-test for unequal variances should be chosen [33]. For this reason, before proceeding with a T-test, the F-test is recommended for testing the variances of two groups with equality [34].

The F-test is named after the statistician R. Fisher [35], with “F” representing the ratio of two variances. In a similar way to the T-test, by comparing this variance ratio “F static” with a value from a table of the f critical variance ratio for a specific significance level, the equality of two variances can be verified. At the 0.05 significance level, the f critical is usually denoted by f_0.05_. Specifically, if the calculated ratio (F) is larger than the f_0.05_ read from the f-Table, the two groups are deemed to have equal variances; otherwise, they have unequal variances.

Table 6 shows the statistical testing results in which the mean value of each group was calculated using data in Table 4 and Table 5, respectively. In the F-test, if the calculated value F static (F) is smaller than the f_critical_ in the 95% confidence interval (f_0.05_), the two populations are deemed to have equal variance with a 95% confidence interval. The mean values of compressive strength, indirect tensile strength and elastic modulus for CFGPC and OPCC samples were accordingly deemed to have equal variance at the α = 0.05 significance level. In the T-test, if a calculated value t static (t_s_) is smaller than the t _critical_ in a 95% confidence interval (t_0.05_), the difference between two populations is then deemed statistically significant with a 95% confidence interval. 

It is thus clear that in the case of comparison between CFGPC and OPCC for their mean of compressive strengths or their mean of indirect tensile strength, statistically significant differences did not exist. In contrast, it could be concluded with a considerably low probability of error that the elastic modulus of CFGPC was significantly lower than that of OPCC. It can be seen that the value of P = 1.4 × 10^−10^ was far less than α = 0.05, which means that the difference in elastic modulus between CFGPC and OPCC had less than a 1.4 × 10^−10^ chance to be a statistical fluke. In other words, the relatively low elastic modulus of CFGPC compared to OPCC was nearly 100% due to the nature of CFGPC.

The elastic modulus (E_c_) describes the capability of concrete samples to resist deformation along the axis of the loads applied. This mechanical characteristic is affected by both aggregates and binder material in the composite [36]. As the aggregates and their volumetric proportions used in the two kinds of concrete CFGPC and OPCC were more or less the same (72% and 75%, respectively), the different elastic modulus results were related to the binders used in these two kinds of concrete. It therefore should be noticed that the elastic modulus of concrete manufactured using class F geopolymer binder should not be predicted using the existing method. 

### 3.4. Correlation between Mechanical Characteristics of CFGPC

Previous practice usually uses the compressive strength of OPCC to predict its elastic modulus and tensile strength [36]. The close correlation between compressive strength and other characteristics has been proven to be possible to define and describe by means of statistical regression. A number of empirical formulae connecting the compressive strength and the other two mechanical characteristics have been proposed in standards.

#### 3.4.1. Correlation between f_t_ and f_c_’ in CFGPC

According to the Australian standard AS3600 [37], the correlation between the tensile and compressive strengths of OPCC is described as
(1)$$\mathrm{AS}3600:\;f_{t}=0.4\sqrt{f_{c}^{\prime }}.$$

In the European code CEB-FIP [38], this relationship is given by
(2)$$\mathrm{CEB}-\mathrm{FIP}:\;f_{t}=0.3\;{f^{\prime }}_{c}^{2/3}.$$

The American Concrete Institution [39] expresses this relationship as
(3)$$\mathrm{ACI}\;363R-92:\;f_{t}=0.59\;\sqrt{f_{c}^{\prime }}.$$

However, as a result of this investigation, it is argued here that a more suitable fitness model may better represent this relationship for CFGPC. 

In this study, the suggested empirical equation representing the relationship between the tensile strength and the compressive strength of CFGPC is determined as
(4)$$f_{t}=0.0876\;f_{c}^{\prime }+0.0585$$
where *f_t_* is the indirect tensile strength, and *f_c_^’^* is the compressive strength of cylinders, all in MPa. 

In deriving this equation, not only the data obtained from the present tests were used, but also those reported by Hardjito [16], Rajini and Rao [40], Chang [41], Ryu et al. [42] and Sofi et al. [43]. In total, 40 pairs of f_c_^’^–f_t_ data sets, including a wide range of strength values, constituted the data pool. The values of the experimental data and the predictions of indirect tensile strength obtained from Equations (1) to (4) are shown in Table 7.

It can be seen that among the four equations, the predicted results calculated from Equation (4) obtained from this study were the closest to the experimental ones. To further illustrate correlations between the tested and predicted results, a scatter plot was used to analysis the compressive versus tensile strength values, and the lines representing the standard equations and the 95% confidence and prediction intervals of Equation (4) are shown in Figure 2. 

The 95% confidence and prediction intervals were calculated according to the approach described by Verschuuren [44]. 

In Figure 2, there are three lines describing the equations suggested by standards that represent the relationship between the f_c_^’^ and f_t_ of OPCC (ACI, AS3600 and CEB-FIP), and one line represents that of CFGPC (Equation (4)-Cui).

All the data points depicted on this figure were collected from research works of CFGPC. It follows from examining these data that the tensile strength values of CFGPC increased with the increase in compressive strength and that these increments kept nearly constant. However, for OPCC, previous research has shown that its tensile strength increased at a decreasing rate with incremental increases in its compressive strength [36,45]. 

Basically, there were two different correlations shown here. As can be seen in Figure 2, the tensile strength of OPCC was proportional to its compressive strength raised to a specific power. In contrast, the correlation between the compressive strength and indirect tensile strength of CFGPC was linear. 

The confidence intervals determined from the experimental data and Equation (4) are represented by the two red curves in Figure 2, marking the limits of the area that with 95% probability the true prediction between CFGPC’s f_t_ and f_c_^’^ should locate [46]. The further away from this area a prediction line was located, the less it would correlate to the true prediction. The prediction interval, which is represented by the yellow lines in Figure 2, marked an area where for each true value of CFGPC’s f_c_^’^, there was a 95% probability that the true value of f_t_ was located within these area.

Of the four equations presented in Figure 2, Equation (1) from AS3600 was the furthest from the confidence and prediction intervals. With increases in strength, this prediction kept deviating from the data points. 

Only short parts of the prediction lines provided by CEB-FIP and ACI 363R-92, Equations (2) and (3), were inside the 95% confidence interval related to the suggested Equation (4). Especially, when the compressive strength continued to increase, these two lines deviated further away from the data points and finally protruded from the bottom lines of confidence and prediction intervals. 

Consequently, none of these three OPCC predictions could describe the correlation between CFGPC’s f_t_ and f_c_^’^. Specifically, the indirect tensile strength values calculated from current provisions were too conservative compared with the real values of CFGPC when the compressive strength was higher than approximately 40 MPa. For another reason, these three OPCC empirical equations overestimated the indirect tensile strength of GPC when their compressive strength was lower than approximately 25 MPa. In fact, the indirect tensile strength values of OPCC and CFGPC only had a short range of commonality and that was when their compressive strength was in the range from 25 to 40 MPa, as illustrated in Figure 2. It is noticed that previous research claimed that CFGPC in general has a higher indirect tensile strength than OPCC [47]. However, according to the results of this study, this conclusion is correct only when the compressive strength of these two concretes is higher than approximately 40 MPa. 

#### 3.4.2. Correlation between f_c_’ and E_c_ in CFGPC

There is hardly any doubt that there must be a correlation between the compressive strength and the elastic modulus of OPCC and CFGPC. Based on substantial data, some empirical equations have been recommended by certain standards for predicting the value of OPCC’s E_c_ from f_c_^’^. For instance, ACI 318-14 Section 19.2.2 [48], recommends the expression for computing the modulus of elasticity for normal-weight OPCC as
(5)$$Ec=4733\sqrt{f_{c}^{\prime }}$$
where *E_c_* is the elastic modulus, and *f_c_’* is the compressive strength, both in MPa.

Diaz et al. [49] proposed a similar formula for CFGPC as
(6)$$Ec=580\;f_{c}^{\prime }$$
where *E_c_* is the elastic modulus, and *f_c_^’^* is the compressive strength of CFGPC, both in MPa.

In this study, a fitness model was obtained from regression to describe the relationship between *f_c_^’^* and *E_c_* of CFGPC as
(7)$$Ec=874.5f_{c}^{\prime }{\;}^{0.85}$$
in which *E_c_* refers to the elastic modulus, and *f_c_^’^* stands for the compressive strength of CFGPC, both in MPa. The data that were used to create this proposed equation included data reported by Fernández-Jiménez et al. [18], Diaz et al. [49] and Hardjito and Rangan [50] together with the data obtained in this research. In total, 35 pairs of data sets were used in developing Equation (7). The experimental and predicted values are shown in Table 8.

It follows from examining the data in Table 8 that the predicted results calculated from Equation (7) (the proposed fitness model) were the closest to the experimental values. The experimental results for the elastic modulus of CFGPC were generally lower than those values calculated from the empirical equations described by the standards of OPCC, such as ACI 318-14, Section 19.2.2 [48]. To further illustrate the correlations between the tested and predicted results, a scatter plot of the compressive versus modulus of elasticity values, and regression lines representing Equations (5)–(7), as well as including the 95% confidence and prediction intervals, are shown in are shown in Figure 3.

As the fitness model investigating the relationship between the compressive strength and elastic modulus of CFGPC was non-linear, the 95% confidence and prediction intervals were obtained according to the approach of linearization suggested by Brown [51]. The complete data pool and detailed calculations may be sourced from Cui [29].

It can be seen that the line representing the equation proposed by the ACI code was far beyond the confidence interval determined according to the experimental data and Equation (7). If the ACI equation were used to predict the E_c_ of CFGPC from specific compressive strength value, the real value would be considerably overestimated. If f_c_^’^ was lower than approximately 25 MPa, the line representing Diaz et al.’s equation would then be situated inside the confidence interval. However, with the increase of f_c_^’^, the Diaz et al. line would protrude outside the interval. The authors have come to the conclusions that, firstly, the current provision applied to OPCC, would if applied to CFGPC, overestimate the elastic modulus of CFGPC at specific compressive strength values. Secondly, the equation suggested by Diaz et al. overestimated CFGPC’s elastic modulus when its compressive strength was higher than 20 MPa. Consequently, it seems that Equation (7), rather than those equations suggested by ACI or Diaz et al., better satisfies prediction of the elastic modulus of CFGPC in the circumstances that only compressive strength data are available.

In Figure 3, it is also observed that some data points obtained from CFGPC samples such as those used by Diaz et al. and this study, were relatively far from the prediction line. This finding indicates that there might be other factors related to the modulus of elasticity of CFGPC other than their compressive strength. Indeed, it is quite well known that the elastic modulus of concrete is influenced by the elastic modulus of aggregates [52]. Moreover, in industrial projects, there are large variations in the densities of produced concretes due to the different aggregates or source materials used. Previous research proved that the density of concrete is also correlated to the elastic modulus [53,54]. Therefore, it is expected that for the CFGPC samples with known density values, the prediction of the elastic modulus will be influenced by the density values. 

#### 3.4.3. Correlation between f_c_’ Concrete Density w and E_c_ in CFGPC

To improve the applicability of predictions, the influence of the density values (w) has been included in many standards [40]. In this study, w was added into the regression as a second predictor apart from the compressive strength, and thus, the regression equation changed to
*E_c_* = 0.35 *f_c_^’^*^2/3^*w*^2^(8)
where *E_c_* is the elastic modulus of CFGPC, in GPa; *w* is the density of CFGPC, in g/cm^3^ and *f_c_^’^* is the compressive strength value, in MPa. The applicability of this equation was supported by including data previously reported by Diaz et al. [49] and listed in Table 8. The plots for Equation (8) at different values of concrete density as well as the experimental data from this study and that of Diaz et al. [49] are shown in Figure 4.

In Figure 4, there are five lines that represent the position of the prediction described by Equation (8). They are the prediction lines when the density of concrete ranged from 1.8 g/cm^3^ to 2.6 g/cm^3^. It is apparent that the elastic modulus of CFGPC is relatively sensitive to the change in density. With the increase of concrete density, the prediction for elastic modulus of CFGPC increased and vice versa.

Additionally, compared with Equation (7), Equation (8) has a higher value of R^2^ (0.8652), which suggests that including density in the prediction contributed to capturing variations in the elastic modulus of CFGPC. Consequently, when both the density and the compressive strength of CFGPC are available, Equation (8) is recommended for quite accurately predicting the elastic modulus.

## 4. Conclusions

This study mainly presents the investigation into three essential mechanical characteristics of fly ash based geopolymer concrete (CFGPC). These are the compressive strength, the indirect tensile strength and the elastic modulus. The experimental results were compared with corresponding ordinary Portland cement concrete (OPCC). The correlations between compressive strength and elastic modulus or indirect tensile strength were determined by statistical regression. The following conclusions can be drawn:

1. It was found that the current provisions used for predicting the indirect tensile strength of OPCC do not accurately apply to CFGPC. It is noticed that the tensile strength of CFGPC increased in an approximately proportional manner with the increase in compressive strength, and the correlation was basically linear, whereas for OPCC, the correlation followed a non-linear relationship. The correlation between the tensile and compressive strength of CFGPC is therefore better expressed using a suggested statistically derived equation, which produced predicted values that preferably compared with those predicted using the current provisions. 

2. The statistical hypothesis testing illustrated that when the compressive strength of the CFGPC mix and the OPCC mix is between 30 MPa and 40 MPa, there is no significant difference in their indirect tensile strength values. However, in this strength range, CFGPC’s mean elastic modulus is statistically significantly lower than that of the OPCC mix. As the aggregates used and their proportion to total mass remained the same for CFGPC and OPCC mixes, the different values of the mean elastic modulus of these two concretes is attributed to the differences in OPC and geopolymer binders. The relevant values for the elastic modulus of CFGPC were found to be significantly lower than those for OPCC even though their compressive strength was equal or higher than that of OPCC. It is therefore likely that the geopolymer binder may be capable of additional elastic deformation compared to the OPC binder.

3. The prediction model suggested for OPCC by ACI 318 significantly overestimated the elastic modulus of CFGPC at each specific strength level. This prediction exceeds the upper boundary of the prediction intervals plotted using the experimental data of CFGPC. Based on that, the authors strongly hold the opinion that using OPCC’s provisions to predict CFGPC’s elastic modulus may not be appropriate. 

4. Using data obtained from testing and the literature, an empirical equation was derived to describe the non-linear correlation between the compressive strength and elastic modulus of CFGPC. The suggested empirical equation is considerably accurate in terms of predicting the elastic modulus when compared with the equation currently used for OPCC. The suggested formula has a reasonably high R^2^ value of 0.7663, and when the variations in density were added in the regression, the validity increased to a higher level of 0.8652.

5. It was proved that including density in the prediction contributed to capturing variations in the elastic modulus of CFGPC. Consequently, when both the density and the compressive strength of CFGPC are available, Equation (8) is recommended for predicting the elastic modulus of CFGPC.

## Figures and Tables

**Figure 1 materials-13-01651-f001:**
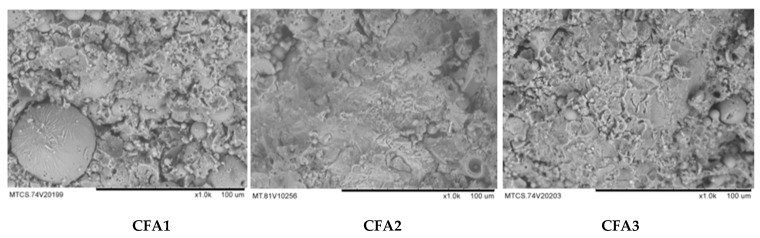
SEM photos for CFGPC made from CFA1, CFA2 and CFA3.

**Figure 2 materials-13-01651-f002:**
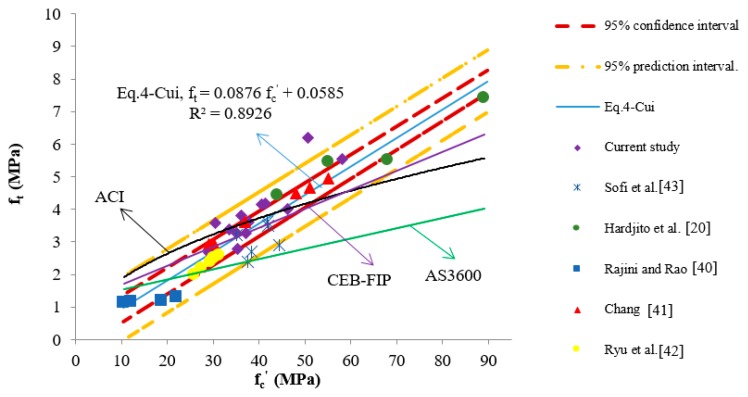
Relationships between splitting tensile strength and compressive strength of CFGPC (AS3600, ACI 363R-92, CEB-FIP, proposed Equation (4), etc.).

**Figure 3 materials-13-01651-f003:**
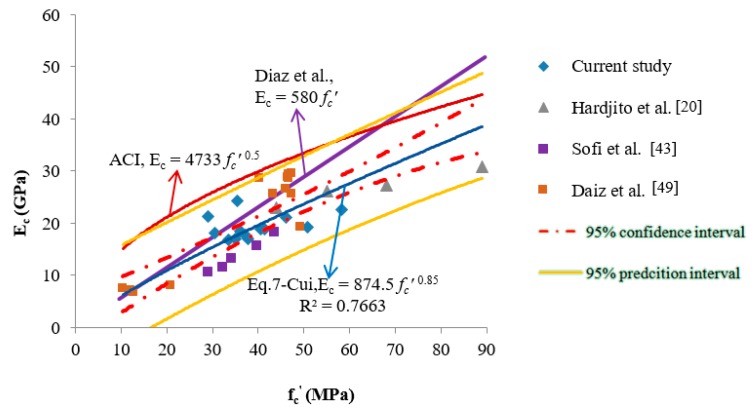
Relationships between elastic modulus and compressive strength of CFGPC (ACI318-14, Diaz et al., proposed Equation (7), etc.)

**Figure 4 materials-13-01651-f004:**
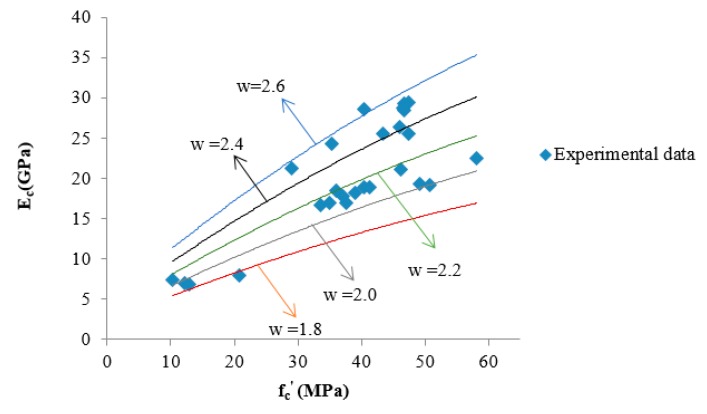
The suggested relationship, Equation (8), between compressive strength and elastic modulus of CFGPC applied with different density values.

**Table 1 materials-13-01651-t001:** Chemical composition of each fly ash as determined by X-ray fluorescence (XRF).

Fly Ash Batches	CFA1	CFA2	CFA3
Component	wt.%	wt.%	wt.%
SiO_2_	58.600	58.491	57.360
Al_2_O_3_	20.202	21.046	22.106
Fe_2_O_3_	9.245	8.286	8.126
CaO	4.670	3.843	4.701
K_2_O	3.023	3.938	3.090
TiO_2_	2.341	2.232	2.445
SO_3_	1.040	1.282	1.098
SrO	0.339	0.340	0.489
ZrO_2_	0.295	0.226	0.263
MnO	0.165	0.158	0.189
Rb_2_O	0.044	0.045	0.053
Y_2_O_3_	0.038	0.032	0.043
Loss of Ignition (AS 3583.3-1991 [22]	0.97	1.6	0.91
SiO_2_/Al_2_O_3_(wt)	2.90	2.78	2.59

**Table 2 materials-13-01651-t002:** Mix quantities per one cubic meter for class F fly ash based geopolymer concrete (CFGPC).

CFGPC 35	Coarse Aggregate			Fine	CFA	NaOH	Na_2_SiO_3_	Free Water	Total
	14 mm	10 mm	7 mm						
Mass (kg/m^3^)	500	310	280	630	420	60	150	31	2389

**Table 3 materials-13-01651-t003:** Mix quantities per one cubic meter, for ordinary Portland cement concrete (OPCC).

OPCC 35	Coarse Aggregate			Fine	Free Water	Cement	Total
	14 mm	10 mm	7 mm				
Mass (kg/m^3^)	242	353	349	814	225	357	2340

**Table 4 materials-13-01651-t004:** Summary of the mechanical properties of CFGPC.

Mixes ID	Concrete Density, w (g/cm^3^)	Compressive Strength, f_c_^’^ (MPa)	Elastic Modulus, E_c_ (GPa)	Indirect Tensile Strength, f_t_ (MPa)	f_t_/f_c_^’^	Fly Ash
1	2.26	35.40	24.29	2.93	8.28%	CFA1
2	2.30	28.99	21.25	2.66	9.18%	CFA1
3	2.24	46.18	21.19	4.00	8.66%	CFA2
4	2.27	36.10	18.47	3.81	10.55%	CFA2
5	2.25	40.48	18.95	4.14	10.23%	CFA2
6	2.25	41.36	18.87	4.19	10.13%	CFA2
7	2.22	33.52	16.74	3.37	10.05%	CFA3
8	2.23	39.07	18.2	3.57	11.70%	CFA3
9	2.24	35.03	16.97	3.28	9.36%	CFA3
10	2.22	37.12	17.78	3.26	8.78%	CFA3
11	2.21	37.69	16.99	3.60	9.55%	CFA3
12	2.22	37.21	17.9	3.65	9.81%	CFA3
Ave	2.24	37.35	18.97	3.54	9.69%	

**Table 5 materials-13-01651-t005:** Summary of the mechanical properties of OPCC.

Sample ID	Concrete Density, w (g/cm^3^)	Compressive Strength, f_c_^’^ (MPa)	Elastic Modulus, E_c_ (GPa)	Indirect Tensile Strength, f_t_ (MPa)	f_t_/f_c_^’^
1	2.36	35.43	28.81	2.97	8.38%
2	2.35	38.88	28.60	3.35	8.62%
3	2.37	39.09	23.12	3.40	8.70%
4	2.35	37.06	28.26	3.40	9.17%
5	2.37	32.96	29.59	2.76	8.37%
6	2.35	34.54	27.62	3.54	10.25%
7	2.34	35.09	28.24	3.42	9.75%
8	2.31	33.96	27.04	2.88	8.48%
9	2.34	36.22	29.12	3.24	8.95%
10	2.33	39.74	29.03	3.82	9.61%
11	2.35	42.30	28.37	3.54	8.37%
12	2.34	37.59	28.91	3.19	8.84%
Ave	2.35	36.91	28.06	3.29	9.01%

**Table 6 materials-13-01651-t006:** Statistical test results for compressive strength of CFGPC and OPCC.

Mechanical Characteristics	f_c_^’^ (MPa)		f_t_ (MPa)		E_c_ (GPa)	
Concrete Type	OPCC	CFGPC	OPCC	CFGPC	OPCC	CFGPC
Mean	36.91	37.35	3.29	3.54	28.06	18.97
Test date	28 day	7 day	28 day	7 day	28 day	7 day
F-test	F = 2.46 < f_0.05_ = 2.81		F = 2.37 < f_0.05_ = 2.81		F = 1.72 < f_0.05_ = 2.81	
Equal variances?	Yes		Yes		Yes	
T-test	t_s_ = 0.30 < t_0.05_ = 2.07 P = 0.77 > 0.05		t_s_ = 1.52 < t_0.05_ = 2.07 P = 0.14 > 0.05		t_s_ = 11.23 > t_0.05_ = 2.07 P = 1.4 × 10^−10^ < 0.05	
Statistically significant difference?	No		No		Yes	

**Table 7 materials-13-01651-t007:** Experimental and predicted values for indirect tensile strength of GPC.

Researcher	Experimental f_t_ (MPa)	Experimental f_c_^’^ (MPa)	AS3600 f_t_ (MPa)	CEB-FIP f_t_ (MPa)	ACI 363 f_t_ (MPa)	Equation (4) f_t_ (MPa)
Current study	5.54	58.2	3.05	4.51	2.14	5.09
	4.82	49.57	2.82	4.05	1.97	4.35
	2.79	35.4	2.38	3.23	1.67	3.14
	2.66	28.99	2.15	2.83	1.51	2.59
	4.00	46.18	2.72	3.86	1.90	4.06
	3.81	36.1	2.40	3.28	1.68	3.20
	4.14	40.48	2.54	3.54	1.78	3.57
	4.19	41.36	2.57	3.59	1.80	3.65
	3.37	33.52	2.32	3.12	1.62	2.98
	3.57	30.5	2.21	2.93	1.55	2.72
	3.28	35.03	2.37	3.21	1.66	3.11
	3.26	37.12	2.44	3.34	1.71	3.29
	3.6	37.69	2.46	3.37	1.72	3.34
Sofi et al. [43]	3.2	35.2	2.37	3.22	1.66	3.12
	2.9	44.4	2.67	3.76	1.87	3.91
	2.4	37.6	2.45	3.37	1.72	3.33
	3.6	41.8	2.59	3.61	1.81	3.69
	3.5	42.0	2.59	3.63	1.81	3.70
	2.7	38.3	2.48	3.41	1.73	3.39
Hardjito [20]	7.43	89	3.77	5.98	2.64	7.71
	5.52	68	3.30	5.00	2.31	5.92
	5.45	55	2.97	4.34	2.08	4.81
	4.43	44	2.65	3.74	1.86	3.87
Rajini and Rao [40]	1.13	10.51	1.30	1.44	0.91	1.02
	1.16	12.11	1.39	1.58	0.97	1.15
	1.18	18.68	1.73	2.11	1.21	1.71
	1.32	22.03	1.88	2.36	1.31	2.00
Chang [41]	3.62	37	2.43	3.33	1.70	3.30
	2.96	30	2.19	2.90	1.53	2.69
	4.96	55	2.97	4.34	2.08	4.88
	4.48	48	2.77	3.96	1.94	4.26
	2.96	30	2.19	2.90	1.53	2.69
	2.93	29	2.15	2.83	1.51	2.60
	4.65	51	2.86	4.13	2.00	4.53
Ryu et al. [42]	2.0	25.8	2.03	2.62	1.42	2.32
	2.2	27.5	2.10	2.73	1.47	2.47
	2.4	29.4	2.17	2.86	1.52	2.63
	2.5	30.3	2.20	2.92	1.54	2.71
	2.5	30	2.19	2.90	1.53	2.69
	2.6	31.2	2.23	2.97	1.56	2.79

**Table 8 materials-13-01651-t008:** Experimental and predicted values for elastic modulus of CFGPC.

Researcher	f_c_^’^ (MPa)	w (g/cm^3^)	E_c_ (GPa)	ACI 318 E_c_ (GPa)	Diaz et al. [49] E_c_ (GPa)	Equation (7) E_c_ (GPa)
Current study	58.20	2.26	22.54	36.11	33.76	27.13
	50.70	2.30	19.24	33.70	29.41	24.15
	35.40	2.24	24.29	28.16	20.53	17.82
	28.99	2.27	21.25	25.48	16.81	15.05
	46.18	2.25	21.19	32.16	26.78	22.31
	36.1	2.25	18.47	28.44	20.94	18.12
	40.48	2.20	18.95	30.11	23.48	19.96
	41.36	2.22	18.87	30.44	23.99	20.33
	33.52	2.22	16.74	27.40	19.44	30.84
	39.07	2.23	18.2	26.14	17.69	17.02
	35.03	2.24	16.97	28.01	20.32	15.71
	37.12	2.22	17.78	28.84	21.53	17.67
	37.69	2.21	16.99	29.06	21.86	18.55
Diaz et al. [49]	40.35	2.31	28.599	30.06	23.40	18.79
	47.35	2.29	29.475	32.57	27.46	19.91
	46.69	2.32	29.358	32.34	27.08	22.79
	46.79	2.31	28.517	32.38	27.14	22.52
	46.11	2.29	26.455	32.14	26.74	22.56
	47.44	2.24	25.635	32.60	27.52	22.28
	12.20	1.99	7.04	16.53	7.08	22.83
	12.82	1.97	6.812	16.95	7.44	7.24
	20.86	1.99	7.96	21.62	12.10	7.55
	10.34	1.89	7.46	15.22	6.00	11.40
	46.56	2.37	28.744	32.30	27.00	6.30
	49.24	1.91	19.278	33.21	28.56	22.47
	43.38	2.29	25.607	31.17	25.16	23.56
Hardjito and Rangan [50]	89		30.8	44.65	51.62	21.16
	68		27.3	39.03	39.44	38.85
	55		26.1	35.10	31.90	30.95
	44		23	31.40	25.52	25.87
Fernández-Jiménez et al. [18]	32		11.7	26.77	18.56	21.42
	29		10.7	25.49	16.82	16.36
	34		13.4	27.60	19.72	15.06
	43.5		18.4	31.22	25.23	17.23
	39.5		15.8	29.75	22.91	21.21

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
