# Peer review of "Experimental and Statistical Study on Mechanical Characteristics of Geopolymer Concrete"

_materials, 2020, doi:10.3390/ma13071651_

Round 1

Reviewer 1 Report

This paper reports the experimental and statistical study on mechanical characteristics of class F fly ash based geopolymer concrete. However, the presented results are not significant and sufficient to be considered for publication in this Journal. Some reasons for rejecting the acceptance of the manuscript to the journal of Materials are given below:

  • The abstract did not explain the summary of the whole research. 
  • The introduction is not enough.
  • The XRD graphs cannot be presented in this way (without clear labeling and no axis title).
  • The presented results and analysis are very preliminary ones.

Reviewer 2 Report

The paper is an experimental and statistical study on mechanical properties of geopolymer concrete. The number of the samples were tested experimentally is good and the statistical models that have been developed based on them, seem to fit quite well with other experimental results from the literature. 

The paper is interesting and can be used for further future research on different geopolymer mix designs. The paper can be considered for publication after some minor corrections which are the following:

Line 100: It is mentioned that superplasticizer is used in the geopolymer mix. More information about the type and the quantity of the superplasticizer is missing.

Line 105: A resting period of 24hrs before oven was used. More explanation why is needed and probably some references.

Line 110: "the heat cured CFGPC can achieve more than 90% strength within 7 days". How do the authors know about that? Have they tested it?

Line 156-177 where the results from the application of the differenstatistical models are given, more explanation of the statistical models (i.e. what is F variance and what is P value) is missing.

Reviewer 3 Report

Good work. Nevertheless, the phase composition and microstructure dependence of mechanical properties for materials should be investigated and commented.

Reviewer 4 Report

This paper discusses the correlation between the mechanical properties of geopolymers. The results confirm that the existing equations for concrete are not suitable to evaluate the mechanical properties of the geopolymer. The paper is well organized and presents valuable discussions.

On the other hand, as the authors also point out in this paper, the mechanical properties of geopolymers are strongly influenced by the mechanical properties of the binder. Not only the Class F fly ash employed in this paper but also various kind of the geopolymer binder are widely employed. Moreover, the mechanical properties of the geopolymer may also depend on the type of alkaline solutions. It is desirable to specify the extent to which the results of this paper are applicable.

In addition to this, I believe this manuscript can be accepted by responding to the comments below.

Tables 2 and 3:

The table only says "14 mm", "10 mm" and "7 mm", but I'm not sure what they mean. They mat be the sizes of the coarse aggregate. Please specify them.

2.6 Curing

The authors say "The CFGPC cylinders were covered with plastic lids immediately after casting to minimize moisture loss." I think the minimization of moisture loss is needed for OPCC as well. Please clarify the common procedures for OPCC and CFGPC.

Line 220, 305:

figure => Figure

Figure 2:

I think the legend for this graph is inappropriate. For example, there is no legend for red triangles or green circles, and the Euro standard doesn't show a line. Also, where authors' names are mentioned, it is better to show references, as in Table 7, etc.

Hardjito [16] and Chang [32] are not mentioned in the legend, but their results seem to be included in the graph.

It is divided into three areas, "Low," "Normal," and "High," but they are not relevant to the discussion. I think this color coding is unnecessary.

Figure 3:

As in Figure 2, please revise the way of the legend and the three divided areas.

Line 321:

"Error! Reference source not found.” appears on the body text. Please correct as appropriate.

Figure 4:

Please indicate the details of the dataset used here. Is it the same dataset as used in Figure 3? In particular, information on density is mandatory.

4. Conclusions:

The conclusion should briefly address the all-important points described in the body text. Here, the discussion of 3.3.3 and Figure 4 is not included.

It is recommended that you summarize them briefly.

Round 2

Reviewer 1 Report

Dear Authors,

Thanks for the corrections and your input data.

This version of the manuscript with a little English improvement can be accepted in the Journal.

Best Regards

Reviewer 3 Report

I accept the corrected paper.